# Air-stable 18-electron adducts of Schrock catalysts with tuned stability constants for spontaneous release of the active species

Henrik Gulyás [1✉], Shigetaka Hayano [2], Ádám Madarász[3], Imre Pápai [3✉], Márk Szabó[1], Ágota Bucsai[1], Eddy Martin[4,5] & Jordi Benet-Buchholz[4]

Schrock alkylidenes are highly versatile, very active olefin metathesis catalysts, but their pronounced sensitivity to air still hinders their applications. Converting them into more robust but inactive 18-electron adducts was suggested previously to facilitate their handling. Generating the active species from the inactive adducts, however, required a high-temperature Lewis acid treatment and resulted in an insoluble by-product, limiting the practicality of the methodology. Herein, we introduce an approach to circumvent the inconvenient, costly, and environmentally taxing activation process. We show that 18-electron adducts of W- and Mo-based Schrock catalysts with finite stability constants (typically $K = 200–15{,}000 \, M^{-1}$) can readily be prepared and isolated in excellent yields. The adducts display enhanced air-stability in the solid state, and in solution they dissociate spontaneously, hence liberating the active alkylidenes without chemical assistance.

[1] XiMo Hungary Ltd, Berlini Park, Budapest, Hungary. [2] Zeon Corporation, Tokyo, Japan. [3] Research Center for Natural Sciences, Institute of Organic Chemistry, Budapest, Hungary. [4] Institute of Chemical Research of Catalonia (ICIQ), Barcelona Institute of Science and Technology (BIST), Tarragona, Spain. [5] Bruker France SAS, Wissembourg, France. ✉email: henrik.gulyas@ximo-inc.com; papai.imre@ttk.hu

Schrock complexes, 14-electron Mo(VI) and W(VI) alkylidenes, are highly efficient catalysts for olefin metathesis (OM) reactions. The understanding of their chemistry has steadily been growing over the course of the last thirty years, leading to the development of newer and newer generations of these remarkable catalysts, and consequently tackling such challenges as the efficient synthesis of macrocyclic olefins[1], enantioselective metatheses[2,3], Z and E selective metatheses[4–7], including the formation of Z-disubstituted enol ethers (cis-G-HC=CH-OR)[8], Z-alkenyl halides (cis-G-HC=CH-X; X = Cl, Br, F)[9], and Z-trifluoromethyl-substituted alkenes (cis-G-HC=CH-CF3)[10]. In addition to the development of the diverse chemistry of Schrock complexes, considerable efforts have also been made to immobilize them[11], successfully addressing the inherent problems of all homogenous catalysts: separation and recycling. Under appropriate conditions, the productivity of these catalysts can also be remarkable. As an example, for one of the latest generations of these d[0] alkylidenes, over 1 million ton has been reported in homo-cross-metathesis of propene[12]. The high activity, amazingly tunable selectivity, and recyclability of these catalysts should allow for widespread applications in both academic research and industrial production. The primary reason why this enormous potential is not yet being fully exploited is the pronounced air sensitivity of these complexes. In comparison, ruthenium-based olefin metathesis catalysts are generally more robust[13,14], which can be the decisive factor in catalyst selection for process development and manufacturing applications[15].

In the past decade, several research groups addressed this problem and developed approaches to facilitate the handling of air-sensitive Schrock alkylidenes. XiMo researchers showed that embedding Schrock alkylidenes into paraffin pellets provided effective physical protection from air[16], while the Fürstner group developed a general approach towards their chemical protection[17,18]. Notably, Fürstner showed that [Mo(NAr^diiPr)(neophylidene)(OCMe(CF3)2)2] (1), a powerful bisalkoxide type Schrock catalyst, readily and quantitatively reacted with 1,10-phenanthroline and 2,2′-bipyridine, and the corresponding 18-electron adducts [Mo(NAr^diiPr)(neophylidene)(OCMe(CF3)2)2(1,10-phenanthroline)] (2) and [Mo(NAr^diiPr)(neophylidene)(OCMe(CF3)2)2(2,2′-bipyridine)] (3) proved largely air-stable (Fig. 1). Although the 18-electron alkylidenes 2 and 3 were not active as olefin metathesis catalysts, the active 14-electron Mo(VI)-alkylidene 1 could be liberated prior to performing the desired OM reactions by the addition of Lewis acidic reagents, such as ZnCl2, which at elevated temperatures was capable of capturing the protecting Lewis base from the coordination sphere of the molybdenum[17] (Fig. 1). The Fürstner group successfully used the same strategy to protect alkylidyne complexes used to catalyze alkyne metatheses[18]. Independently from Fürstner's findings, Schrock and Hoveyda showed that the bipyridine adducts of highly unstable molybdenum imido alkylidene bispyrrolide complexes displayed enhanced chemical and air stability. The authors also had to use ZnCl2 to liberate the bispyrrolides for the in situ syntheses of Schrock catalysts[19].

Although Fürstner elegantly and convincingly demonstrated that Lewis base complexation has a great potential to protect Schrock catalysts in air, his approach has not yet gained widespread use. One of the possible reasons for this is the necessity of the high-temperature Lewis acid treatment to liberate the active 14-electron species from the corresponding inactive adducts, generating an insoluble by-product in the process. The chemical activation might only be a mere inconvenience on laboratory scale, but it could represent major technical difficulties, extra cost,

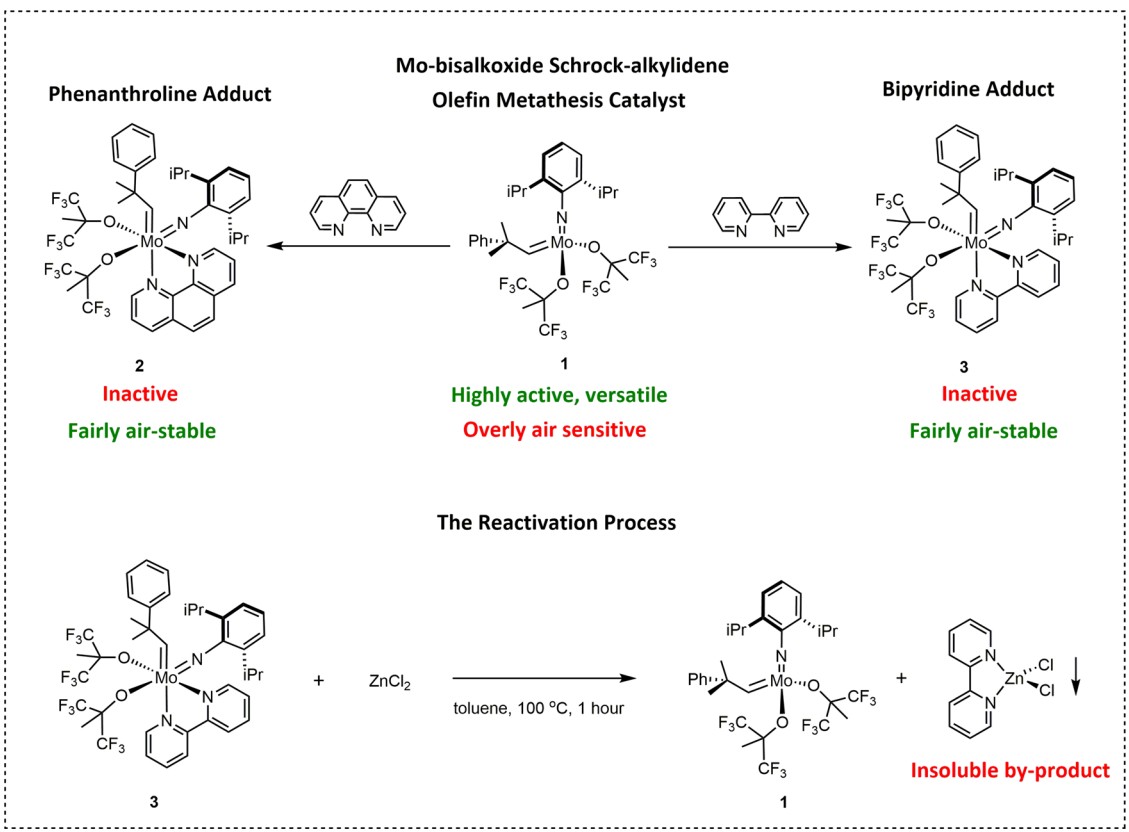

**Fig. 1 Fürstner's approach towards air-stable 18-electron adducts of Mo-bisalkoxide Schrock alkylidenes[17].** The Mo-bisalkoxide Schrock alkylidene [Mo(NAr^diiPr)(neophylidene)(OCMe(CF3)2)2] (1) readily and irreversibly reacts with both 1,10-phenanthroline and 2,2′-bipyridine, yielding the air-stable but inactive 18-electron adducts 2 and 3; the active catalyst 1 can be liberated from 2 and 3 via a high-temperature treatment with appropriate Lewis acids.

and undesirable environmental impact in industrial-scale production.

Fürstner's study focused exclusively on one specific class of Schrock catalysts, molybdenum-bisalkoxide-alkylidenes. Due to the weak sigma donor capacity of the alkoxide ligands of the chosen Schrock catalysts, strong coordination of the protecting Lewis bases occurred independently of the structural differences of the applied bidentate N-heterocyclic ligands, hence, the necessity of the activation process.

Examining a greater scope of Schrock alkylidenes, involving both molybdenum- and tungsten-based catalysts in both bisalkoxide and mono alkoxide pyrrolide (MAP) ligand environments, we observed that the thermodynamic stability of their adducts could readily be influenced by the structure of the Lewis base. Based on this observation, we envisioned circumventing the use of activating agents simply by adjusting the stability constants of the adducts so that the active species are spontaneously released in solution.

## Results and discussion
### On the thermodynamic stability of bipyridine and phenanthroline adducts of Schrock alkylidenes.

In Fig. 2 we show typical examples of how the structural differences of the applied Lewis bases allow for adjusting the thermodynamic stability of the 18-electron adducts of Schrock catalysts.

1,10-Phenanthroline readily reacted with both W(VI)-disiloxide catalyst 4 and W(VI) MAP catalyst 5, and complete conversions were observed within an hour. From both the $C_s$ symmetric 4 and the asymmetric 5 one single geometric isomer was formed, and the respective 18-electron adducts 6 and 7 could be isolated in high yields (b, Fig. 2; Supplementary Figs. 1–3).

Similarly to Fürstner's phenanthroline and bipyridine protected Mo(VI)-alkylidenes 2 and 3, the coordinatively saturated 6 and 7 showed markedly increased air stability with respect to their 14-electron parent complexes: 6 proved bench-stable for weeks, while the MAP-derived 7 with the small and strongly electronegative alkoxide ligand for several hours. It is worth noting that 5, the parent complex of 7, is an extremely sensitive

Schrock catalyst. In 10-mg quantities, it decomposes within minutes upon exposure to air, in sharp contrast with its phenanthroline adduct 7. Considering catalytic applications, 6 and 7 were inactive in olefin metatheses, but they released the active parent complexes upon treating them with appropriate Lewis acids, such as ZnCl₂·dioxane. In short, 6 and 7 are typical examples of Fürstner-type adducts.

In contrast, when choosing 2,2′-bipyridine as the Lewis base, the reactions with 4 and 5 do not progress to completion; rather, they result in equilibrium mixtures between the bipyridine adducts 8, 9 and the corresponding 14-electron W(VI) complexes (a, Fig. 2).

It is worth noting that the lower stability constants of 8 and 9, compared to the phenanthroline adducts 6 and 7, can readily be rationalized based on the distinctive structural features of the corresponding Lewis bases. 1,10-Phenanthroline is a planar, structurally rigid, $C_{2v}$ symmetric molecule, in which the relative position of the two N-donor atoms is ideal for chelate formation (A, Fig. 3). In the structurally similar 2,2′-bipyridine, the pyridyl rings can rotate around the C2-C2′ sigma bond. As demonstrated by DFT calculations (Supplementary Figs. 15 and 16; Supplementary Table 8), the lowest-energy conformation of 2,2′-bipyridine is the $C_{2h}$ symmetric, coplanar s-trans structure, which clearly does not favor chelate formation, or coordination in general (B, Fig. 3). In this anti-coplanar conformation, intramolecular hydrogen bonds between each N-atom and the ortho-hydrogen of the adjacent pyridyl ring provide significant stabilization.

The $C_{2v}$ symmetric syn-coplanar conformation C, analogous to the structure of 1,10-phenanthroline, is predicted to be +6.6 kcal·mol⁻¹ less stable than B. This conformation has proven to be a transition state on the potential energy surface rather than an energy minimum. The notable destabilization of the $C_{2v}$ structure is due to the Coulomb repulsion between the nonbonding pairs, an unfavorable intramolecular dipole–dipole interaction between the pyridyl rings, and the steric repulsion between the ortho-hydrogens (C, Fig. 3). We should also note that the most stable syn conformations of the molecule have ±40° torsion angles (D,

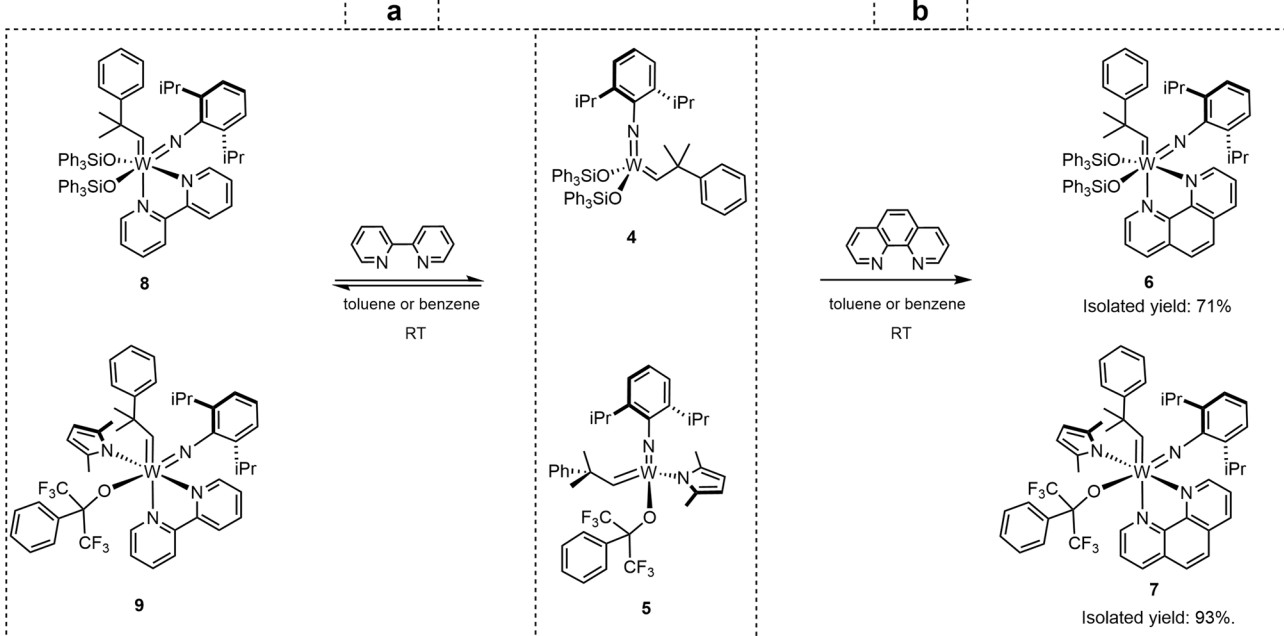

**Fig. 2 Reactions of W(VI)-disiloxide catalyst 4 and W(VI) MAP catalyst 5 with 1,10-phenanthroline and 2,2′-bipyridine.** Both 4 and 5 react with 1,10-phenanthroline quantitatively and irreversibly (b), while their reactions with 2,2′-bipyridine are reversible, lead to equilibrium mixtures (a).

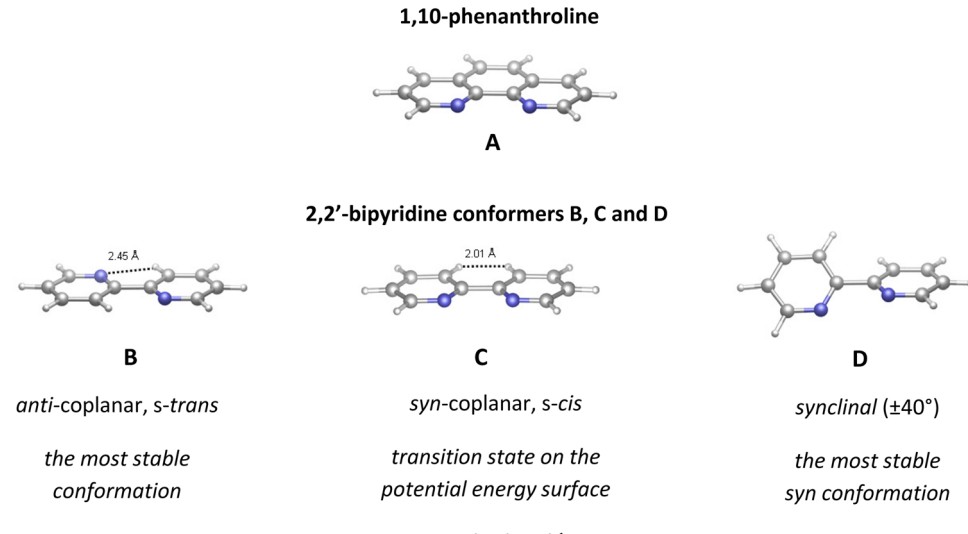

**1,10-phenanthroline**

**A**

**2,2'-bipyridine conformers B, C and D**

2.45 Å                                          2.01 Å

**B**                                    **C**                                    **D**

*anti*-coplanar, s-*trans*          *syn*-coplanar, s-*cis*          *synclinal* (±40°)

*the most stable*                  *transition state on the*          *the most stable*
*conformation*                   *potential energy surface*          *syn conformation*

0.0 kcal·mol$^{-1}$              +6.6 kcal·mol$^{-1}$              +4.5 kcal·mol$^{-1}$

**Fig. 3 Comparison of the structure of 1,10-phenanthroline and relevant conformers of 2,2'-bipyridine.** The optimized structures of 1,10-phenanthroline (**A**) and the relevant 2,2'-bipyridine conformers (**B**, **C**, **D**). The corresponding relative Gibbs free energies have been obtained from DFT calculations performed at the $\omega$B97XD/Def2TZVPP//$\omega$B97XD/Def2SVP// level.

Fig. 3). The Gibbs free energy of these *synclinal* conformations is still +4.5 kcal·mol$^{-1}$ higher than that of the *anti*-coplanar conformation **B**. The decreased stability constants of the bipyridine adducts **8** and **9**, as compared to those of **6** and **7**, are therefore related to the strain induced in the *syn*-coplanar structure required for the coordination of the ligand.

**18-Electron adducts of Schrock alkylidenes with finite stability constants: synthesis and solution-phase behavior.** One of the key points to our proposal is that having an equilibrium between a Schrock catalyst and a Lewis base does not pose any difficulty or disadvantage from a synthetic point of view. The labile adducts **8** and **9** could readily be isolated by shifting the equilibria towards the adduct formation.

In the case of **8**, we rapidly found a combination of conditions under which the adduct started to crystallize from the reaction mixture spontaneously. The crystallization removes **8** from the liquid-phase equilibrium, hence prompting the desired shift towards the adduct formation, and eventually resulting in excellent isolated yield (91%) (Fig. 4; Supplementary Fig. 4). In the case of **9**, we reasoned that in an $M + L \rightleftharpoons ML$ equilibrium, at a [M]:[L] = 1:1 stoichiometry, increasing the concentration by evaporating the solvent will inherently drive the equilibrium towards the clean formation of ML, as [ML] squarely increases with respect to the increase of [M] and [L]. Indeed, combining in situ formed **5** with one equivalent of bipyridine and then evaporating the solvent yielded crude **9** as an orange-red solid. The dimethylpyrrole by-product could readily be removed by the trituration of the solids with pentane. The simple process yielded **9** as a bright orange solid in 83% overall yield (Fig. 4; Supplementary Figs. 6 and 7). It is notable that the parent compound **5**, a yellow crystalline solid, is notoriously difficult to purify. It is difficult to initiate its crystallization and the removal of the dimethylpyrrole is also challenging. The isolated yields typically vary between 45 and 55%, in sharp contrast with the rapid, facile, and efficient isolation of adduct **9**.

As concentrating the solution of the 14-electron Schrock alkylidenes and one equivalent of bipyridine drove the equilibria towards the formation of the corresponding adducts, dissolving the adducts and diluting the solutions inherently result in shifting the equilibria towards dissociation. This means that in the case of

these labile 18-electron adducts, the liberation of the catalytically active 14-electron species should not require any chemical assistance, a distinct feature that **2**, **3** or **6** and **7** do not possess. Indeed, the NMR spectra of these complexes showed a very clear concentration dependence, in line with this expectation. Figure 5 depicts the alkylidene range of the $^{1}$H NMR spectra of **9**, as well as the corresponding $^{19}$F NMR spectra. The spectra were recorded at 0.05, 0.01, and 0.002 M concentrations. Despite the equilibrium, the signals of the different species are sharp, well-separated, and do not shift significantly. Both the alkylidene signals in the $^{1}$H NMR spectra [**5**: 9.24 (s), **9**: 12.04 ppm (s)] and the signals of the diastereotopic CF$_3$ groups in the $^{19}$F NMR spectra [**5**: –72.7 (q), –74.6 ppm (q); **9**: –75.3 (broad q), –69.0 ppm (q)] can serve to determine the ratio of the 14-electron and 18-electron species (Fig. 5). The spectra demonstrate how the active 14-electron complex **5** is released upon diluting the solution of the storage complex **9**. Even in the highly concentrated 0.05 M C$_6$D$_6$ solution of **9**, 24% of the adduct dissociates and liberates **5** (blue spectra in Fig. 5). Upon decreasing the tungsten concentrations to 0.01 M, the ratio of the liberated 14-electron species increased to 45%, and when diluting the C$_6$D$_6$ solution to a [W] = 0.002 M concentration the ratio of **5** reached 72%. From the results of the NMR analyses, the stability constant of **9** was estimated to be $K_{9, \text{C6D6, 298K}} = 270\ \text{M}^{-1}$. The stability constant of **8** was calculated similarly as $K_{8, \text{C6D6, 298K}} = 14000\ \text{M}^{-1}$.

Despite the labile nature of the adducts **8** and **9**, they could be fully characterized not only in solution but also in the solid phase. Crystals suitable for single-crystal X-ray diffraction could readily be obtained from **8** as the adduct spontaneously crystallizes from the reaction mixture under the optimized conditions. The MAP-derived adduct **9** was crystallized from a toluene solution by slow evaporation of the solvent (Fig. 6; Supplementary Figs. 5 and 8; Supplementary Tables 1–6; Supplementary Data 1 and 2). The solid-state structures of the bipyridine adducts **8** and **9** are analogous to the structure we have suggested for the corresponding phenanthroline adducts **6** and **7**. The arylimido ligands are *trans* to the alkoxide/siloxide ligand, which defines unequivocally the stereostructure of the pseudo-octahedral complexes. The crystals of both **8** and **9** were racemates, they contained both enantiomers of the chiral structures, as evidenced by the crystals' centrosymmetric space group $P2_1/c$.

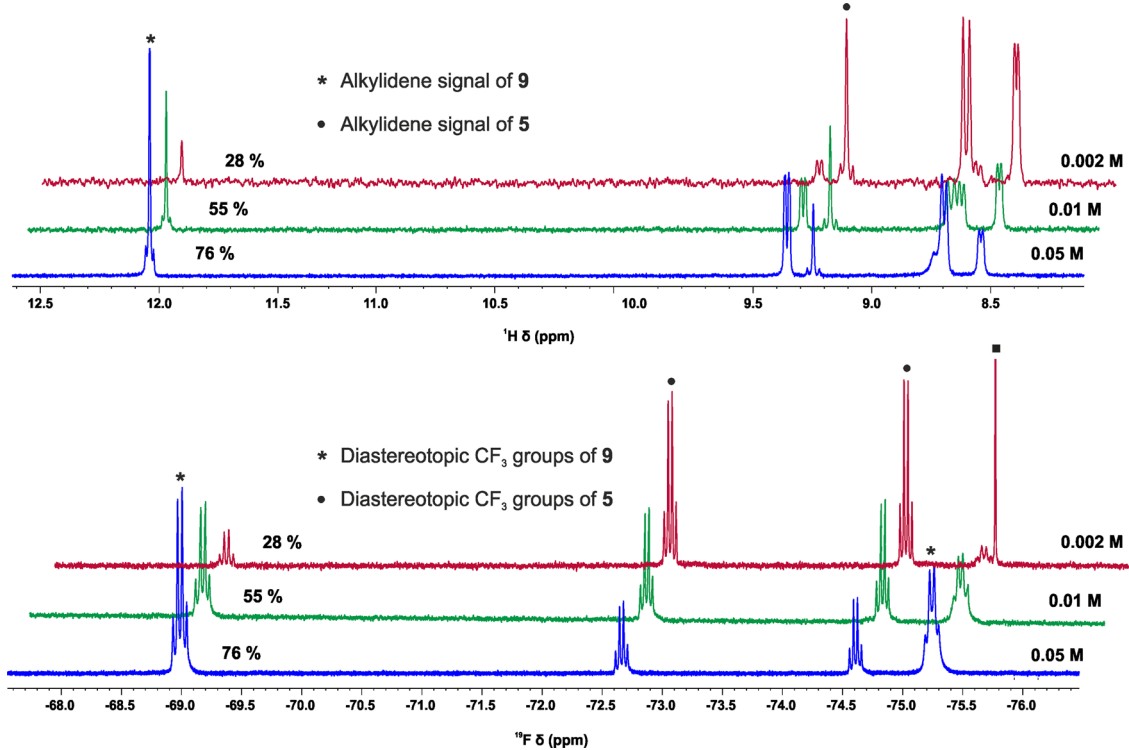

**Fig. 4 Isolation of the labile 18-electron adducts 8 and 9.** The equilibria can readily be shifted towards the adduct formation either by in situ crystallization of the adduct, or by concentrating the reaction mixture, allowing the isolation of **8** and **9** in excellent yields.

**Fig. 5 ¹H NMR spectra (alkylidene range) and ¹⁹F NMR spectra of 9 recorded at 25 °C, in C₆D₆, at varying concentrations.** Concentrations: 0.05 M (blue), 0.01 M (green) and 0.002 M (red). The black square depicts Ph(CF₃)₂COH derived from slight decomposition.

Theoretical calculations of the Gibbs free energies of the complexation of **5** with 1,10-phenanthroline and 2,2′-bipyridine provided additional insights into the thermodynamic differences between the formation of the Fürstner-type adduct **7** and its labile analog **9**. The solution-phase Gibbs free energies were calculated using the density functional theory at the ωB97XD/Def2TZVPP// ωB97XD/Def2SVP level, in benzene as the solvent. The obtained results are in line with our experimental observations (Fig. 7; Supplementary Fig. 17; Supplementary Tables 8 and 9). The formation of **7** has proved unambiguously exergonic. The Gibbs free energy change of complexation with 1,10-phenanthroline is predicted to be $-8.5\,\text{kcal·mol}^{-1}$, indicating that the adduct formation is clearly favored thermodynamically. With bipyridine as the Lewis base, the $\Delta G$ of the adduct formation is considerably

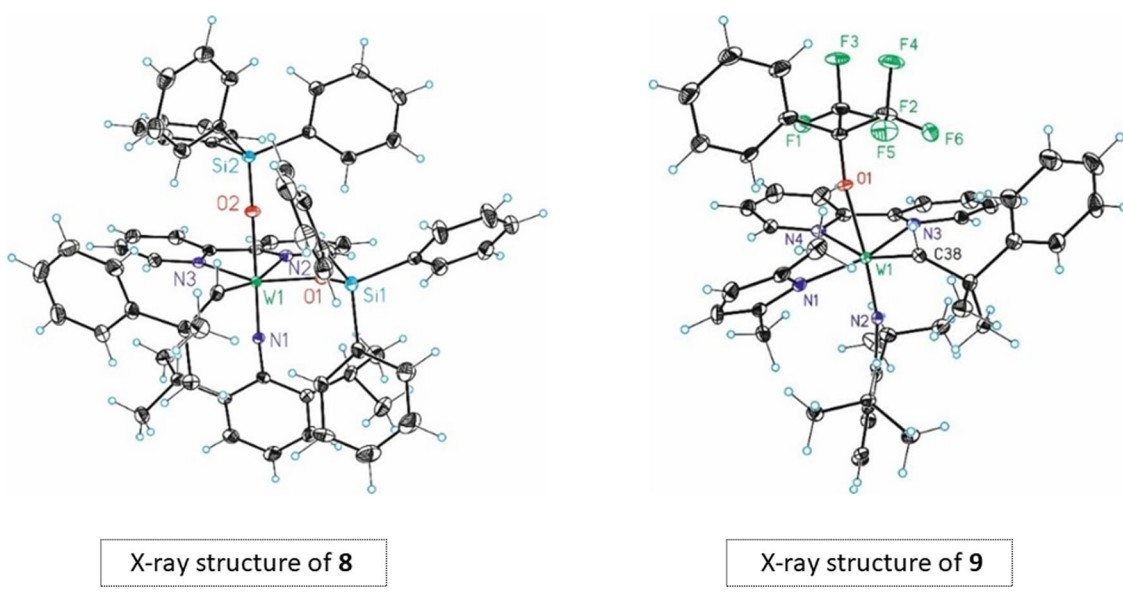

X-ray structure of 8                    X-ray structure of 9

**Fig. 6 Structures of the bipyridine adducts 8 and 9 in the solid state.** Both structures have distorted octahedral geometry. In both structures, the arylimido ligands are *trans* to the alkoxide/siloxide ligand, and the neophylidene ligands are *syn*-oriented (oriented towards the imido ligand).

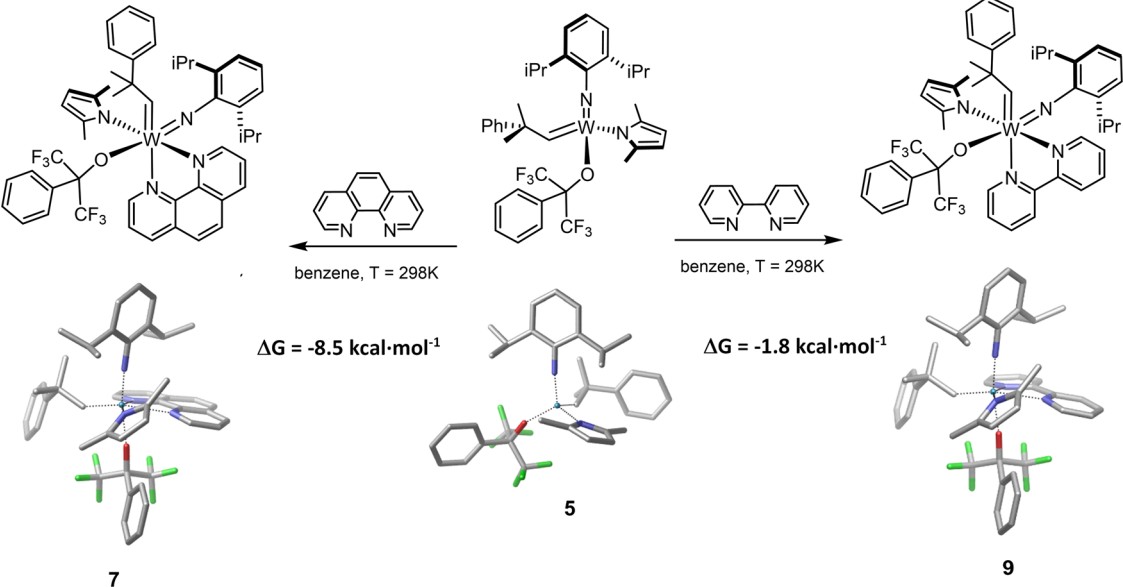

**Fig. 7 DFT calculations on the Gibbs free energies of complexation of 5 by 1,10-phenanthroline and 2,2'-bipyridine.** Conditions: ωB97XD/Def2TZVPP// ωB97XD/Def2SVP level; in benzene as solvent (SMD, Solvation Model based on Density); at 298 K; with thermal and BSSE corrections.

higher, $-1.8\,\mathrm{kcal \cdot mol^{-1}}$, suggesting an association–dissociation equilibrium between the Schrock alkylidene and the bidentate N-heterocycle. It is worth noting that the experimentally observed $K_{9,C6D6,298K} = 270\,\mathrm{M^{-1}}$ binding constant translates to $\Delta G = -3.3\,\mathrm{kcal \cdot mol^{-1}}$ Gibbs free energy change of association, in reasonable agreement with the DFT predictions.

Other thermodynamic parameters of the adduct formation, among them the standard enthalpy in particular, also have great practical importance in terms of controlling the degree of dissociation. Since the association of the bipyridine to the 14-electron complex **5** should inherently have an entropy cost, the coordination can be expected to be exothermic, taking into account the slightly negative Gibbs free energy of the reaction. An exothermic association means that increasing the temperature will promote the dissociation of the adduct (Le Chatelier's principle). Indeed, we found that increasing the temperature of

the solution of **9** did result in the expected decrease of the stability constant, as shown in Fig. 8. The NMR experiments displayed in Fig. 8 were performed in $C_6D_6$, at a $[W] = 0.05\,\mathrm{M}$ concentration, and van't Hoff analysis of the data gave $\Delta H = -25.2\,\mathrm{kcal \cdot mol^{-1}}$ and $\Delta S = -73.5\,\mathrm{cal \cdot mol^{-1} \cdot K^{-1}}$ as the enthalpy and the entropy of the association, respectively, in line with our original assumption. In a control set of experiments, we also performed the van't Hoff analysis at a $[W] = 0.01\,\mathrm{M}$ concentration. The control experiments essentially provided the same thermodynamic parameters for the conversion of **5** to **9** (Fig. 8; Supplementary Figs. 13 and 14; Supplementary Table 7). It is of practical importance that complete (>99%) dissociation can be achieved even in the highly concentrated 0.05 M benzene solution before the temperature reaches 70 °C.

It is noteworthy that the air stability of the labile adducts **8** and **9** does not differ from that of the Fürstner-type **6** and **7**. The

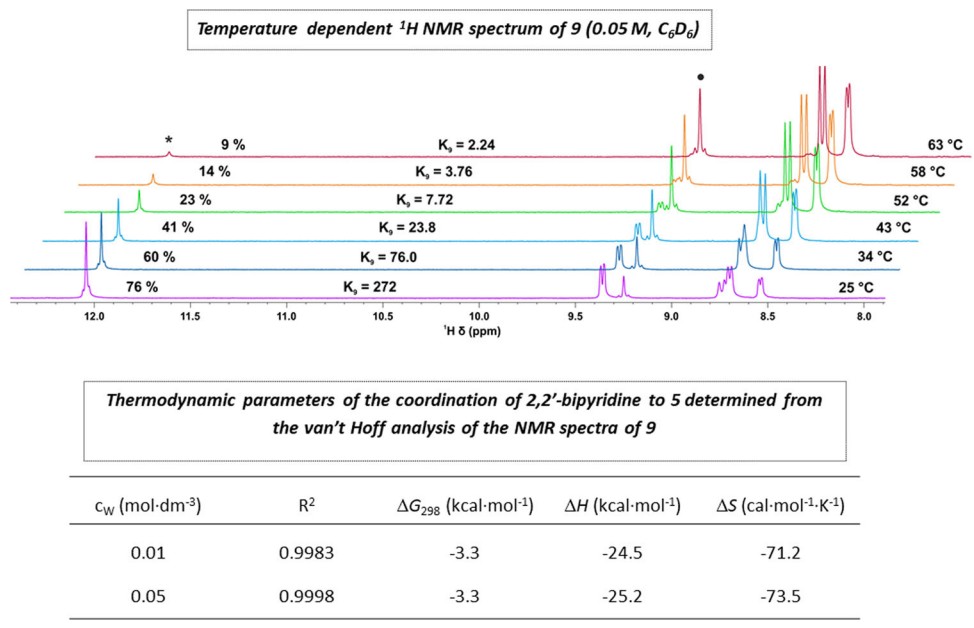

**Thermodynamic parameters of the coordination of 2,2′-bipyridine to 5 determined from the van't Hoff analysis of the NMR spectra of 9**

| $c_W$ (mol·dm$^{-3}$) | $R^2$ | $\Delta G_{298}$ (kcal·mol$^{-1}$) | $\Delta H$ (kcal·mol$^{-1}$) | $\Delta S$ (cal·mol$^{-1}$·K$^{-1}$) |
|---|---|---|---|---|
| 0.01 | 0.9983 | -3.3 | -24.5 | -71.2 |
| 0.05 | 0.9998 | -3.3 | -25.2 | -73.5 |

**Fig. 8 Temperature dependence of the $^1$H NMR spectrum of 9 (0.05 M, C$_6$D$_6$) and thermodynamic parameters of the complexation of 5 by 2,2′-bipyridine derived from the van't Hoff analysis of the NMR spectra of 9.** Conditions: 0.01 M and 0.05 M solutions of **9** were analyzed by $^1$H and $^{19}$F NMR in the 298 K–323 K temperature range (five different temperatures) and in the 298 K–336 K temperature range (six different temperatures), respectively; the corresponding equilibrium constants were determined based on the integrals of the alkylidene and CF$_3$ signals of **5** and **9**; the thermodynamic parameters were determined from the $R·\ln K$ vs. $10^3·T^{-1}$ van't Hoff plots.

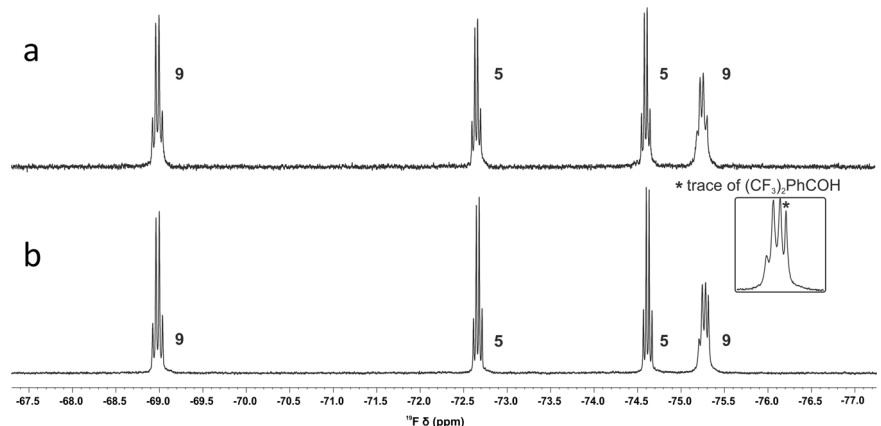

**Fig. 9 Enhanced air stability of the labile adduct 9.** $^{19}$F NMR spectra of **9** (0.01 mmol) kept in the glove box (**a**) and left on the bench in the air for 2 h (**b**). NMR conditions: [W] = 0.01 M, C$_6$D$_6$, 298 K.

tungsten-bisalkoxide adduct **8** proved air-stable for weeks, displaying basically identical robustness to that of its non-labile analog **6**. The tungsten-MAP-derived **9** proved more sensitive, but similarly to its phenanthroline-chelated counterpart **7**, it remained stable in air for several hours (Fig. 9). Considering that **5** decomposes within minutes upon exposure to air, it is clear that the labile adducts **8** and **9** are not only easier to synthesize and isolate, but also much more facile to handle, requiring less strict conditions compared to the corresponding 14-electron alkylidenes.

In terms of the synthetic scope of the approach, we should emphasize that the stability constant of these adducts can readily be modified via the steric and electronic tuning of the protecting bidentate Lewis bases. For instance, a more stable storage complex for **5** could easily be prepared using 5,5′-dimethyl–2,2′-bipyridine, an inexpensive bipyridine derivative with increased basicity (Fig. 10; Supplementary Figs. 9 and 10). We observed a binding constant of $K_{10, \text{C6D6, 298K}} = 1700 \text{ M}^{-1}$ for **10**. Comparison of the stability

constants of **10** and **9** ($K_{9, \text{C6D6, 298K}} = 270 \text{ M}^{-1}$) shows that the simple structural modification of the Lewis base increased the stability of the storage complex of **5** by an order of magnitude, despite the fact that the electron-donating methyl groups are bound to the 5-C and 5′-C carbon atoms of the bipyridine, hence increasing slightly its steric bulk, too.

The present approach can readily be applied to molybdenum alkylidenes, as well. In fact, Mo(VI)-alkylidenes are less Lewis acidic than W(VI)-alkylidenes in the same ligand environment; therefore, the coordination of any Lewis base to a molybdenum-based Schrock catalyst will be weaker than to its tungsten-based analog. As an important example, Fürstner's molybdenum-based inactive adduct **3** could easily be rendered labile by replacing bipyridine with a weaker sigma donor derivative, 4,4′-dibromo-2,2′-bipyridine. Unlike the Fürstner adducts **2** and **3**, **11** spontaneously dissociates in solution, releasing the 14-electron Schrock complex **1** without any chemical assistance. We found that the binding constant of **11** was $K_{11, \text{C6D6, 298K}} = 9000 \text{ M}^{-1}$ in

**Fig. 10 Tuning of the stability constant of the 18-electron storage complexes via tuning the basicity of the Lewis base.** Using Lewis bases substituted with electron-donating or electron-withdrawing groups can increase or decrease the stability constant of the adducts, respectively, hence extending the scope of the approach.

deutero-benzene, at 298 K (Fig. 10; Supplementary Figs. 11 and 12).

It is worth noting that there is an ample pool of substituted bidentate N-heterocycles that are commercially available, and generally inexpensive, offering a readily accessible tool to control the binding constant of such adducts independently of the nature of the metal and the ligand environment of the 14-electron alkylidenes.

The adducts **10** and **11** also showed markedly enhanced air stability compared to their parent compounds. Notably, we observed negligible decomposition in 0.01-mmol samples of **11** left in air for several hours. Based on $^{19}$F NMR analyses, ca. 2% of **11** decomposed after 2 h, and ca. 5% after 24 h, while under identical conditions the yellow crystals of **1** turned black within 20 min, and $^{1}$H and $^{19}$F NMR analyses showed quantitative decomposition of the 14-electron alkylidene.

**Catalytic behavior of the labile adducts**. Although extensive catalyst and substrate screenings are beyond the scope of this work, a few comparative metathesis studies will highlight the similarities and differences in the catalytic behavior of the adducts compared to that of the parent complexes.

Most importantly, as our results on the solution-phase equilibria of **8–11** may suggest, these adducts directly serve as precatalysts in olefin metathesis reactions. Unlike the phenanthroline adducts **2**, **6** and **7**, or the bipyridine adduct **3**, they do not require chemical activation.

Under appropriate conditions, the labile adducts will provide very similar catalytic performance to that of the parent complexes. For instance, in the homo-cross-metathesis of methyl oleate with **1** and **11**, performed at 0.05 mol% or 0.02 mol% catalyst loading, and at 80 °C, within 4 h 90% conversions were observed for both the bisalkoxide catalyst and its 18-electron

adduct, and the product distributions were also comparable (Table 1; Supplementary Figs. 21 and 22).

Another important characteristic of these novel precatalysts is that their catalytic behavior may also differ from that of the corresponding parent complex. The differences can mostly be attributed to the interaction of the Lewis base with the parent complex, and/or with the 14-electron and 16-electron species of the catalytic cycle derived from the parent complex. As a typical example, in ring-closing metathesis of diethyl diallyl malonate, we observed a tempered activity for **11** with respect to that of **1**, at room temperature, at a [W] = 0.002 M. In this set of experiments, the catalyst concentration was kept constant, while the substrate: tungsten ratio was gradually increased from 200 (0.5 mol% catalyst) to 2500 (0.04 mol% catalyst). The total reaction time was 24 hours in all cases, but each reaction mixture was sampled and analyzed at 1, 4, and 24 h. (Supplementary Figs. 18–20) Under the applied conditions, **1** showed high activity: complete conversion was obtained within 4 h in each case (Entries 2, 4, 6, 8; Table 2). In contrast, **11** displayed decreasing initial turnover frequencies; the conversion observed at 1 h dropped from 58 to 13% as the substrate:tungsten ratio was increased from 200 to 2500 (Entries 1, 3, 5, 7; Table 2). Nonetheless, the slower reaction rate did not prevent the 18-electron storage complex from providing full conversions within 24 h.

In this specific case, one of the factors that must be considered as an explanation for the difference between the catalytic performances of **1** and **11** is the partial dissociation of the adduct under the catalytic conditions. In benzene solution, at 25 °C, the stability constant of **11** has been found to be ca. 9000 $M^{-1}$. Consequently, at room temperature and 0.002 M tungsten concentration, only ca. 20% of the **11** will dissociate and liberate **1**.

Furthermore, we note that partial dissociation is not the only factor that can alter the catalytic performance of the adducts with respect to their parent complexes. Comparing the 14-electron MAP complex **5** and its bipyridine adduct **9** in ROMP of norbornene, we also experienced a substantially slower reaction in the case of the adduct, despite the fact that based on the stability constant of **9**, and the reaction conditions, over 80% of **5** could be expected to be liberated in the reaction mixture (Table 3; Supplementary Table 10; Supplementary Figs. 23–34). In the reactions shown in Table 3, the catalyst loading was 0.5 mol%, and the tungsten concentration was set to 0.001 M. The reactions were performed in toluene, at 25 °C, and they were monitored by GC using cyclooctane as the internal standard. Reaching full conversion required only 5 h when the 14-electron MAP complex **5** was applied as the catalyst, while it took 5 days in the case of the bipyridine adduct **9**. To understand the substantial difference in the activities, one should consider how the presence of the

---

**Table 1 Homo-cross-metathesis of methyl oleate[a].**

| Entry | Catalyst | Sub./Cat | Conversion | 9OD Yield | 9OD Z/E ratio | OD 9ODDAME Yield | ME 9ODDAME Z/E ratio | ME Yield |
|---|---|---|---|---|---|---|---|---|
| 1 | **1** | 2000 | 90% | 51% | 19/81 | 42% | 19/81 | 48% |
| 2 | **11** | 2000 | 90% | 51% | 20/80 | 49% | 19/81 | 41% |
| 3 | **1** | 5000 | 90% | 52% | 20/80 | 48% | 19/81 | 42% |
| 4 | **11** | 5000 | 89% | 51% | 20/80 | 41% | 20/80 | 48% |

[a]Methyl oleate: 1 mmol, **1** or **11**: 50 μL or 20 μL 0.01 M benzene stock solution (0.0005 mmol or 0.0002 mmol), stirred at 80 °C for 4 h. The reaction mixture was analyzed by GC-MS.

---

**Table 2 Ring-closing metathesis of diethyl diallyl malonate[a].**

| | | | Yield[b] | | |
|---|---|---|---|---|---|
| Entry | Catalyst | Substrate/Catalyst | 1 h | 4 h | 24 h |
| 1 | **11** | 200[c] | 58% | 97% | >99% |
| 2 | **1** | 200[c] | >99% | - | - |
| 3 | **11** | 500[c] | 49% | 95% | >99% |
| 4 | **1** | 500[c] | >99% | - | - |
| 5 | **11** | 1000[c] | 27% | 77% | 99% |
| 6 | **1** | 1000[c] | 99% | >99% | - |
| 7 | **11** | 2500[d] | 13% | 54% | 98% |
| 8 | **1** | 2500[d] | 99% | >99% | - |

[a]Diethyl diallyl malonate: 0.4—5 mmol, **1** or **11**: 200 μL 0.01 M benzene solution (0.002 mmol), stirred at room temperature for 24 h in an open 5 mL vial. The reaction mixture was analyzed at 1, 4, and 24 h. [b]GC yield. [c]The reaction mixture was diluted to 1 mL total volume with benzene. [d]Neat.

**Table 3 Ring-opening-metathesis polymerization of norbornene[a].**

| Entry | Catalyst | Conversion | Time to full conversion | cis:trans ratio | racemo:meso ratio[b] |
|---|---|---|---|---|---|
| 1 | **5** | >99% | 5 hours | 89:11 | 86:14 |
| 2 | **9** | >99% | 5 days | 96:4 | 100:0 |

[a]ROMP in toluene at 25 °C; [NB] = 0.2 M, [1-octene] = 0.04 M; [**5** or **9**] = 0.001 M, [NB]/[1-octene]/ [**5** or **9**] = 200/40/1 mol/mol/mol; monomer conversions were monitored using GC; cyclooctane was employed as an internal standard for NB. [b]Determined for the hydrogenated poly(NB)s.

bipyridine in the reaction mixture may affect the possible interactions of the substrate with the catalyst. Although the equilibrium between **5** and **9** is shifted towards the dissociation of the adduct, the liberated Lewis base will still compete with the substrate for coordination site on the propagating species. Both our experimental and computational studies have shown that the coordination of 2,2′-bipyridine to **5** is slightly exergonic (−3.3 kcal·mol⁻¹). According to the literature, and our findings, the π-coordination of the olefin to the supposedly 14-electron propagating species can be expected to be endergonic[20]. The difference between the Gibbs free energies of the olefin coordination and that of the bipyridine can be as high as 20 kcal·mol⁻¹. Since the bipyridine coordination is much more favored than the coordination of the substrate, the presence of the N-donor ligand in the system can hinder the coordination of the substrate even at relatively high olefin:bipyridine ratios.

Another evidence that the dissociated bipyridine interferes with the catalytic cycle is the dramatic difference between the stereoselectivities of the two precatalysts. The conventional 14-electron MAP complex **5** provided modest *cis* selectivity and racemo:meso ratio. Since the alkoxide ligand of **5** is relatively small, while its arylimido ligand is relatively large, this result is in good agreement with the model Hoveyda and Schrock established for the *cis* selectivity of MAP-type catalysts in different olefin metathesis reactions[4,5]. In sharp contrast, the novel adduct **9** displayed outstanding stereospecificity: it yielded 96% *cis*-, syndiotactic polynorbornene.

These results imply that the active species–Lewis base equilibrium can be used as an additional tool to influence stereoselectivity in metatheses towards specific geometric isomers, diastereoisomers, or enantiomers.

## Conclusions

In conclusion, we have demonstrated that 14-electron Schrock alkylidenes can be converted into 18-electron adducts with finite stability constants using readily available bidentate N-heterocyclic Lewis bases. The stability constant of the adducts can be controlled via the electronic and steric tuning of the Lewis bases. The synthesis of these coordinatively saturated labile "storage complexes" is facile and high yielding, and oftentimes it does not require the isolation of their parent complexes. In such cases, the overall yields of the syntheses considerably exceed the yields of the parent complexes.

The finite stability constants still ensure increased air stability in the solid state, similarly to the increased robustness of the inactive adducts; however, the catalytically active parent complexes can be liberated from the labile 18-electron alkylidenes in solution without the use of Lewis acids or other chemical assistance, successfully circumventing the undesirable activation process and eliminating the resulting insoluble by-product.

The thermodynamic parameters of the adduct formation offer an ample toolbox to control the equilibrium, which is particularly important in terms of the catalytic applications. The protecting Lewis bases may or may not interfere with the catalytic cycle, depending on the adduct, the reaction, and the reaction conditions. Importantly, such interactions offer an additional tool to influence the outcome of the metathesis reactions. We have highlighted this via a highly stereoselective ROMP of norbornene initiated with adduct **9**. The outstanding stereospecificity of **9** is in sharp contrast with the modest stereoselectivity induced by the parent complex **5** induces under the same conditions.

The approach is simple, general, and economical. It offers significant advantages in terms of catalyst synthesis, catalyst stability, and catalytic applications; hence, it should further advance the use of Schrock alkylidenes in academic laboratories and industrial applications alike.

## Methods

**Synthesis and characterization of adducts 6-11, theoretical calculations, and catalytic experiments**. See Supplementary Information for all experimental details. For the NMR spectra of **6–11**, see Supplementary Figs. 1–4, 6, 7, 9–12. For the data on the X-ray analyses of **8** and **9**, see Supplementary Figs. 5 and 8, Supplementary Data 1 and 2, and Supplementary Tables 1–6. For details of the van't Hoff analyses, see Supplementary Figs. 13, 14, and Supplementary Table 7. For the details of the theoretical calculations, see Supplementary Figs. 15–17, and Supplementary Tables 8 and 9. For the details of the catalytic experiments, see Supplementary Figs. 18–34, and Supplementary Table 10.

**Typical synthetic procedure for adducts with finite stability constants; the synthesis of 9**. The bispyrrolide precursor W(CHCMe₂Ph)(NArᵈⁱⁱᴾʳ)(Me₂Pyr)₂ (Arᵈⁱⁱᴾʳ = 2,6-diisopropylphenyl, Me₂Pyr = 2,5-dimethylpyrrolide) (170 mg, 0.25 mmol) was dissolved in toluene (3 mL). Ph(CF₃)₂COH (61 mg, 42 μL, 0.25 mmol) was added. The residues from the vial of the alcohol were rinsed into the reaction vial with toluene (2 × 1 mL). The reaction mixture was stirred for 3 h at room temperature. An aliquot of the reaction mixture was analyzed by ¹H and ¹⁹F NMR. Both methods indicated complete conversion into the desired MAP complex. 2,2′-Bipyridine (39 mg, 0.25 mmol) was added as a solid. The reaction mixture turned dark orange immediately. The residues from the vial of the bipyridine were rinsed into the reaction vial with toluene (2 × 1 mL). The reaction mixture was stirred for 20 min at room temperature, and then the solvent was slowly evaporated in vacuo. The resulting dark orange solids were washed with pentane. The solids were isolated by filtration and washed with small amounts of cold pentane. The product was dried first in vacuum induced N₂ stream, and then in a high vacuum. Orange solid. Yield: 205 mg (83%).

**Catalytic example, homo-cross-metathesis of methyl oleate**. In a glove box, purified methyl oleate (339 μL, 296 mg, 1 mmol) was transferred into an oven-dried 4-mL vial by an automatic pipette. A stock solution of the catalyst, **1** or **11** (0.01 M in benzene, 50 μL, 0.0005 mmol) was added to the substrate. The vial was sealed, and the reaction mixture was stirred at 80°C for 4 hours. The reaction mixture was quenched with MeOH (100 μL). A sample was taken (2 μL), it was diluted with DCM (1 mL), and the solution was analyzed with GC.

## Data availability

All experimental procedures and analytical data (¹H, ¹⁹F, ¹³C NMR, high-resolution mass spectrometry and crystallographic data) can be found in the Supplementary Information. Crystallographic data for compounds **8** (CCDC: 2021603) and **9** (CCDC: 2021604) can be downloaded free of charge from the Cambridge Crystallographic Data Centre (www.ccdc.cam.ac.uk), or accessed from Supplementary Data 1 and 2 of this article. All details of the theoretical calculations can be found in the Supplementary Information. Upon reasonable request, additional data that further support the findings of this study can be provided by the authors.

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

## Acknowledgements

We dedicate this work to the memory of Prof. Georg Fráter, XiMo's co-founder, founding board member, former Chief Executive Officer, Chief Operating Officer, and Chief Scientific Officer; his spirit, scientific curiosity, and leadership are greatly missed. We thank Dr. Zsófia Dubrovay, Dr. Irene Maijó Ferré and Dr. Sònia Abelló Cros for their contribution to the spectroscopic characterizations of adducts 6−11, and Mrs. Ildikó Bodó for her skillful technical assistance in the catalytic experiments. We are grateful to Dr. Levente Ondi and Prof. Georg Fráter for fruitful discussions and their active support in the course of the development of this work. We thank Prof. Amir Hoveyda for his constructive comments on the manuscript prior to submission.

## Author contributions

H.G. proposed the concept and synthesized all tungsten and molybdenum complexes. Á.M. and I.P. performed the computational studies. M.S. performed the NMR characterization of the tungsten and molybdenum complexes and carried out the van't Hoff analyses. E.M. and J.B.B. are responsible for the X-ray characterization of **8** and **9**. S.H., Á.B. and H.G. performed the catalytic experiments. H.G. and I.P. wrote the first version of the manuscript with the contribution of all authors. All authors contributed to the final version of the manuscript.

## Competing interests

The authors M.S., Á.B. and H.G. are researchers at XiMo Hungary, a company that commercializes Schrock alkylidenes. H.G. holds the patent EP3268377B1, WO2016142849A1, which includes in the claims the synthesis of some of the alkylidene complexes described in the manuscript. S.H., Á.M., I.P., E.M. and J.B.B. declare no competing interests.
