## [Peer Review File · Communications Chemistry]

Reviewers' comments:

Reviewer #1 (Remarks to the Author):

Schrock's alkylidene complexes are established catalysts for olefin metathesis. The most active complexes of this type, however, are extremely air and moisture sensitive. An approach to circumvent this problem is addressed in the present work, although the main idea is largely based on previously published work by the groups of A. Fürstner and R. R. Schrock. As mentioned in the Introduction, Fürstner has shown that alkylidene complexes can be stabilized by the addition of chelating Lewis bases such as phenanthroline or bipyridine. Although the resulting complexes are air resistant or even permanently air stable, they do not promote olefin metathesis, and activation with Lewis acids such as metal halides is required. The practicality of this procedure is limited and affords an insoluble by-product.

Using the same strategy, the authors report here the coordination of phenanthroline (phen) and bipyridine (bipy) to two different tungsten alkylidene complexes, which resulted in the formation of complexes 6–9 with enhanced air stability in the solid state. In contrast to the phen complexes 6 and 7, the bipy complexes 8 and 9 show dissociation of the bipy ligand in solution, releasing the catalytically active 14-electron species, which is in equilibrium with the bipy complexes. Remarkably, the authors describe the appropriate conditions to shift the equilibrium into the desired direction. Thus, the chelated complexes can be readily isolated upon crystallization or concentration of the reaction mixture. In contrast, the complexes dissociate spontaneously in (highly) diluted solutions, liberating the active species. Alternatively, full dissociation of the Lewis base is achieved by increasing the temperature. Based on NMR analyses, the authors accurately determined the equilibrium constant (or stability constant) for the formation of adducts 8 and 9. Additionally, thermodynamic parameters were determined for the formation of complex 9. These values agree with the experimental observations. DFT calculations also support the experimental data.

Of particular interest is the fact that the association/dissociation equilibrium can be controlled by adjusting the electronic and steric properties of the Lewis base or by the nature of the parent complex (e.g. W vs. Mo). This may indeed affect the catalytic behavior, and the presence of the free ligand may hinder the coordination of the substrate. In comparative metathesis studies, complex 11 performs similarly to parent complex 1, however, it should be noted that the reactions proceed significantly slower with the bipy complex, revealing that the dissociated bipy ligand acts as a catalyst poison. The same observation is made for the ROMP reaction, which requires 5 days in the presence of 9 in contrast to 5 hours in the presence of 5. Interestingly, however, the stereoselectivity increases for the ROMP catalyzed by the "poisoned" catalyst.

Overall, the method described in this contribution will certainly catch the attention of scientist working in the same field. Despite the small substrate scope, I recommend publication of this nice paper, since it will certainly help to overcome the obstacles of applying Schrock-type olefin catalysts more broadly. The paper is technically sound and should be published after the minor corrections listed below:

Minor corrections:

Page 2

- General: It should be noted that similar strategies, i.e. phen and bipy coordination, have been employed for the stabilization molybdenum alkylidene complexes as alkyne metathesis pre-catalysts.
- Opposed to Schrock's complexes, the family of Grubbs' catalysts contain several examples of robust and air stable complexes. It would be desirable to mention this in the introduction and include pertinent references.
- line 31: please set "Z" and "E" in italics.

- line 37: the zero in "d0" should be superscript.
- lines 43, 45, 46: please enclose the complexes in brackets.

Page 7:

NMR studies: was the formation of the anti isomer observed after the bipyridine ligand has been released? (See <https://doi.org/10.1021/om00052a033>)

Line 180: please use minus signs for negative values (in place of hyphens). Same applies on page 10.

Pages 10 and 11: please use italics for symbols of energies G and H, and enthalpy S.

Page 14, line 345: the phrase "stirred at room temperature for 24 hours in an open 5 mL vial." is out of place.

Reviewer #2 (Remarks to the Author):

Despite their excellent selectivities and activities, Mo- and W-based catalysts for olefin metathesis have not gained the same widespread academic and industrial use as the user-friendly Ru-based systems, mainly due to their inherent air- and moisture sensitivity.

Previously, to make these W and Mo-complexes longer-time stable in air, the packaging into paraffin pellets has been invented (Frater et al., *Org. Process Res. Dev.* 2016, 20, 1709–1716).

The authors described a very clever method to make the title Schrock alkylidenes more stable, thus much more user-friendly.

The problem is that Fürstner et al. described a very similar idea nine years ago (*J. Heppekauser, A. Fürstner, Angew. Chem.* 2011, 123, 7975-7978; *Angew. Chem. Int. Ed.*, 2011, 50, 7829-7832). In fact, like the authors, he reacted Schrock-type bisalkoxide catalysts with 2,2'-bipyridine and 1,10-phenanthroline leading to octahedral complexes that are stable in air over several weeks. To regenerate the parent compounds, these complexes required activation by treatment with anhydrous ZnCl₂ in toluene at up to 100 °C for 30 minutes.

While in the present manuscript dissolution/concentration and heating/cooling are used to shift the equilibrium and as a result to liberate or arrest the active species.

One can argue that this offers a more elegant solution for the re-creation of active Schrock-type bisalkoxide catalysts, because of the hygroscopic nature of ZnCl₂ the original Fürstner process is slightly less user-friendly. But this is only a small difference!

Also, during activation, an air- and moisture sensitive Schrock catalyst is reformed, so the solvent, the protecting gas (not air) and the glassware and substrates must be anyway highly pure, I presume.

As a practitioner in organometallic chemistry having an industrial background, I like the present improvement very much. The paper reads well. Phys-chem part is solid and scholarly made. But I do not think that it has that level of novelty that required for papers published in *Communications Chemistry* journal from Springer Nature group.

After all, this is only a small technical improvement over Fürstner's discovery that shall be published in *OPRD* or *Aldrichimica* type journal gaining the well-deserved visibility and practical use.

Therefore, I cannot recommend acceptance.

The manuscript COMMSCHEM-20-0387-T by Henrik Gulyás et al. presents a synthetic strategy to stabilize Schrock type carbenes with the aim of favoring its handling and application. The resulting precatalysts are 18 e species containing a chelating base that can be removed by controlling the reaction conditions. The strategy is not completely new as first examples were published before by Fürstner and co-workers. However, authors tuned the based and catalyst properties to prevent the need of additional Lewis acids to achieve the precatalyst activation. Thus, in my opinion the improvement is significant and merits publication. Moreover, authors showed the versatility of the approach by using different alkylidene complexes as well as different Lewis basis so that the properties of the precatalysts can be tuned to fit the requirements of the olefin metathesis process under study. Finally, authors showed that the presence of the Lewis base slows down the reaction (as expected) but also it has an influence on the catalyst selectivity. Overall, I believe that the manuscript should be accepted in Communications Chemistry once the three following remarks have been taken into account.

First, authors claim that *“The present approach can readily be applied to molybdenum-alkylidenes, as well. In fact, Mo(VI)-alkylidenes are less Lewis acidic than W(VI)-alkylidenes in the same ligand environment, therefore, the coordination of any Lewis base to a molybdenum-based Schrock catalyst will be weaker than to its 300 tungsten-based analog.”* However, the unique new example that they report is a subtle variation of the Fürstner pre-catalyst. I think authors should justify their statement with a larger list of examples and particularly adding some MAP species. I wonder if the fact that Mo-alkylidenes are less acidic than the tungsten analogues may not imply that the coordination of the base is too weak and thus no 18e adduct is formed for Mo MAP alkylidenes and the Lewis bases used along the work.

Secondly, authors should at least justify the catalyst choice in each reaction. I would have expected that authors explored the same set of catalyst for each reaction (**1**, **5**, **9** and **11**) to see if similar effects are observed when stabilizing different 14 e complexes in similar reaction conditions. However, authors only compare either **1** and **11** or **5** and **9**, thus reader cannot identify if the observed effects are universal or case specific (lower activities/ higher stereoselectivities).

Finally, authors claim that the Lewis base influences the stereoselectivity of Norbornene polymerization. This effect cannot be explained by the model developed by Schrock and

Hoveyda regarding the steric of the imido and alkoxide ligands (ligands in cis- to the metallacyclobutane), except the Lewis base remains coordinated through one nitrogen along the reaction. It seems to me that other effects are more plausible, but I must admit that I find these results quite surprising. In this context, I have the impression that an explanation or some experimental evidences of what is taking place during norbornene polymerization with **11** will largely increase the impact of the present work.

Dear Reviewers,

We are thankful for your careful reading and analysis of our manuscript, for your appreciation, and for your constructive proposals to improve the paper. Upon the Editor's kind invitation, now we have revised the manuscript in line with your comments. In the revised version, we have directly implemented as many of the proposals as the goal and scope of the paper made possible. Also, in this letter, we respond to your comments point by point.

Reviewer 1

We are particularly grateful to Reviewer 1 for the meticulous reading and analysis of the manuscript.

The first general comment of the reviewer is that

„It should be noted that similar strategies, i.e. phen and bipy coordination, have been employed for the stabilization molybdenum alkylidyne complexes as alkyne metathesis pre-catalysts.”

Indeed, this advance in alkyne metathesis is directly related to the concept Fürstner developed for the chemical protection of Schrock-type olefin metathesis catalysts. We introduced a note on this in the manuscript with reference to Fürstner's paper Heppekausen, J., Stade, R., Goddard, R., & Fürstner, A. Practical New Silyloxy-Based Alkyne Metathesis Catalysts with Optimized Activity and Selectivity Profiles. *J. Am. Chem. Soc.* **132**, 11045–11057 (2010).

The reviewer also comments that

„Opposed to Schrock's complexes, the family of Grubbs' catalysts contain several examples of robust and air stable complexes. It would be desirable to mention this in the introduction and include pertinent references.”

This is an important point. In the realm of well-defined olefin metathesis catalysts, ruthenium-alkylidenes are more robust generally, and they can be favored over Schrock alkylidenes as a consequence, including when a catalyst is considered for an industrial application. We have noted these facts in the revised version of the manuscript, and referenced corresponding review articles. [Vougioukalakis, G. C. & Grubbs, R. H. Ruthenium-Based Heterocyclic Carbene-Coordinated Olefin Metathesis Catalysts. *Chem. Rev.* **110**, 1746–1787, (2010). Piola, L, Nahra, F & Nolan, S. P. Olefin metathesis in air. *Beilstein J. Org. Chem.* **11**, 2038–2056, (2015). Carolyn S. Higman, C. S., Lummiss, J. A. M & Fogg D. E. Olefin Metathesis at the Dawn of Implementation in Pharmaceutical and Specialty-Chemicals Manufacturing. *Angew. Chem. Int. Ed.* **55**, 2 –16 (2016).]

The reviewer has a question in relation to our NMR studies on the labile adduct **9**.

“Page7:

NMR studies: was the formation of the *anti* isomer observed after the bipyridine ligand has been released? (See <https://doi.org/10.1021/om00052a033>)”

By NMR, we have not observed the *anti* isomer in the case of either the 14-electron catalyst complexes or the 18-electron adducts. On the other hand, we do think that the *anti* isomers could be present in small quantities in solution, and it is quite probable that the catalyst-Lewis base equilibrium has an impact on the *syn-anti* interconversion. Notably, such an effect could be the reason why the presence of the Lewis base influences the stereoselectivity in NB ROMP catalyzed by **9**. As Prof. Schrock remarks in *J. Am. Chem. Soc.* **131**, 7962–7963 (2009): “Previous ROMP studies suggest that *anti* species may be orders of magnitude more reactive than *syn* species and that trans C=C bonds can form even though no *anti* alkylidene can be observed”. In this context, it is also worth pointing out that the solid phase structure of **9** is the *syn* stereoisomer.

Reviewer 1 has also pointed out some errors in the text of technical or typographical nature. All of them have been corrected in line with his/her recommendations.

Reviewer 2

First of all, we thank the reviewer for his/her notes of appreciation. The reviewer has not voiced any specific concern in relation to the manuscript, but commented on (i) the novelty of the approach in general, and (ii) on the fact that the 14-electron catalyst released upon dissociation of the adduct is still a very air- and moisture-sensitive species. We will briefly address these general concerns.

The idea of the concept presented in the manuscript certainly originates from Fürstner’s discovery of the increased air-stability of 18-electron Schrock catalyst adducts. However, we do not think that tuning the stability constants of the adducts towards the spontaneous release of the active species in solution is an obvious step forward. In fact, the Fürstner group seems to have considered the possible equilibria between Schrock alkylidenes and Lewis bases as a drawback, and not as a feature they could capitalize on: “*In contrast to the known adducts of Schrock alkylidenes with various monodentate donor ligands (ethers, phosphines, pyridine, quinuclidine), in which the coordination of the base is partially reversible in solution at ambient temperature,[15] the NMR spectra of 2, 3, 5, and 7 give no indications for partial dissociation of the chelating bipyridine or phenanthroline residues.[16]*” (Heppekausen, J. & Fürstner A. Rendering Schrock-type Molybdenum Alkylidene Complexes Air Stable: User-Friendly Precatalysts for Alkene Metathesis. *Angew. Chem. Int. Ed.* **50**, 7829 –7832 (2011).) Therefore, using the catalyst – Lewis base equilibria as a switch between the 14-electron active species

and the 18-electron protected adduct, and tuning the stability constants of the adducts to achieve this are certainly original ideas.

The second concern of the reviewer is a very interesting one. Fürstner showed that phenanthroline and bipyridine adducts of Schrock catalysts are considerably more robust, both in the solid phase and in solution, than the 14-electron parent complexes. The increased air-stability facilitates the synthesis, isolation, storage, transportation of these precatalysts. However, the high-temperature Lewis acid activation irreversibly liberates the sensitive active species, and as a consequence, in the course of the catalytic application the Fürstner approach does not offer extra protection. The physical protection of Schrock catalysts in the form of polyolefin wax pellets, developed at XiMo some years ago and referenced by the reviewer (Ondi, L, Nagy, G. M., Czirok, J. B., Bucsay, A., Frater, G. E. From Box to Bench: Air-Stable Molybdenum Catalyst Tablets for Everyday Use in Olefin Metathesis. *Org. Process Res. Dev.*, **20**, 10, 1709–1716, (2016)), has a similar issue. It offers reasonable air-stability in the solid pellets, but as soon as the pellets are dissolved in the reaction mixture, nothing protects the active species. Our present approach is slightly different, and it does offer some extra protection in catalytic applications, as well, since due to the equilibrium a portion of the catalyst is present in the system in the form of the more robust adduct. Importantly, the equilibrium also acts as a buffer in the decomposition process. As the decomposition proceeds, gradually eliminating the active 14-electron catalyst, the corresponding shift in the equilibrium will be triggered, resulting in the dissociation of the adduct and the release of the active species.

In the introduction section of the paper, we have briefly introduced the approach to the physical protection of Schrock alkylidenes described in *Org. Process Res. Dev.*, **20**, 10, 1709–1716, (2016), and referenced by the reviewer.

Reviewer 3

The comments of the third reviewer also showed great insight into the present approach. We address each point below.

In relation to the specific recommendations of the reviewer, first, we should note that the labile adducts presented in the paper were carefully chosen from our catalyst library to demonstrate the most characteristic synthetic and/or application aspects of the proposed approach. They represent the synthetic diversity in terms of both the central atom (W: **8**, **9**; Mo: **11**) and the ligand environment (bisalkoxide: **8**, **11**; MAP: **9**); they demonstrate the possible strategies towards isolation (**8**: crystallization; **9**: solvent evaporation); and they serve to exemplify the most typical catalytic behaviors the user may encounter (little to no difference between the parent catalyst and the adduct,

tempered activity of the adduct due to the presence of the Lewis base, effect of the Lewis base on stereoselectivity). Incidentally, Fürstner in his seminal paper *Angew. Chem. Int. Ed.* **50**, 7829–7832 (2011) also used three 14-electron Schrock alkylidenes to demonstrate his concept, but all three of them were molybdenum-bisalkoxide-type catalysts.

The first specific concern of the reviewer is related to our statement that Mo-alkylidenes are less Lewis acidic than W-alkylidenes in the same ligand environment. We pointed this out in our paper to emphasize that 14-electron Mo-alkylidenes can also readily form labile complexes with the right choice of the Lewis base, and we demonstrated this by “converting” the most thermodynamically stable adducts Prof. Fürstner published (**2** and **3**) into a labile, “autoactivating” derivative (**11**). The necessity of the statement and the chosen example derives from the fact that Prof. Fürstner only presented non-labile Mo-bisalkoxide-alkylidene adducts in his paper and the corresponding patent WO 2012/116695 A1, which might suggest to the non-specialized professionals that the 14-electron Mo-bisalkoxide Schrock catalysts inherently and inevitably form such stable adducts with N-heterocyclic Lewis bases. The fact that we have successfully generated a labile adduct from Mo(NAr^{diiPr})(neophylidene)(OCMe(CF₃)₂)₂ (**1**) bearing a strongly electropositive Mo center implies that Fürstner’s other examples with bulkier and less electronegative ligands are even more favorably disposed towards the formation of adducts with finite stability constants. Notice, for instance, that Fürstner has published Mo(NAr^{diiPr})(neophylidene)(OSiPh₃)₂(1,10-phenanthroline) as a non-labile adduct. In accordance with his result and with the fact that tungsten(VI) complexes are more Lewis acidic than the corresponding Mo(VI) complexes, we show in our paper that W(NAr^{diiPr})(neophylidene)(OSiPh₃)₂(1,10-phenanthroline) (**6**) is non-labile, too. On the other hand, we also show that the corresponding 2,2’-bipyridine adduct **8** is labile, which of course means that the bipyridine analog of Fürstner’s inactive Mo(NAr^{diiPr})(neophylidene)(OSiPh₃)₂(1,10-phenanthroline) is also labile. Importantly, the reviewer rightly concludes that molybdenum MAP catalysts are the least Lewis acidic 14-electron Schrock complexes within the general (and of course oversimplified) structure matrix the two metals and the bisalkoxide or MAP ligand systems define. However, they are far from being out of reach. To give a reference point, please consider that the stability constant of W(NAr^{diiPr})(neophylidene)(Me₂Pyr)(OCPh(CF₃)₂)(2,2’-bipyridine) (**9**) is $K_{9, C_6D_6, 298K} = 270 M^{-1}$, as shown in the manuscript. Under the same conditions, the stability constant of Mo(NAr^{diiPr})(neophylidene)(Me₂Pyr)(OCPh(CF₃)₂)(2,2’-bipyridine) is $K_{9Mo, C_6D_6, 298K} = 7 M^{-1}$, further adjustable readily to more stable but still labile adducts using the approach we demonstrated by comparing W(NAr^{diiPr})(neophylidene)(Me₂Pyr)(OCPh(CF₃)₂)(2,2’-bipyridine) (**9**) and W(NAr^{diiPr})(neophylidene)(OCPh(CF₃)₂)(5,5’-dimethyl-2,2’-bipyridine) (**10**). (Stronger Lewis bases 4,4’-dimethyl-2,2’-bipyridine and 4,4’-dimethoxy-2,2’-bipyridine are also available commercially.) Finally,

we must point out that the examples we present in the paper not only define adequately the scope of the approach we propose, but they also demonstrate a greater scope of Prof. Fürstner's approach (see **6** and **7**) compared to the examples published by the Fürstner group. Therefore, we think that the examples given in the paper sufficiently define the scope of the approach.

The second proposal of the Reviewer is to compare all labile adducts in all catalytic examples. As we have pointed out above, the catalyst examples have not been selected to demonstrate some kind of trend in the catalytic behavior. As a consequence, it would be hard to justify in the paper what we want to achieve with such a comparison, and it will certainly not generate any new insight into the proposed approach.

Finally, Reviewer 3 also points out that the observed effect of the W-MAP adduct **9** on the stereoselectivity of NB ROMP, compared to the stereoselectivity the parent complex W-MAP **5** induces under the same conditions, cannot be explained by the Schrock-Hoveyda model published in *J. Am. Chem. Soc.* **131**, 7962–7963 (2009), and argues that giving a plausible explanation could considerably increase the impact of our work. Although we do agree with the reviewer on both points, at the moment we cannot provide an evidence-based interpretation of this interesting phenomenon. Currently, we believe that the labile Lewis bases might affect the *syn:anti* ratio in favor of the *syn* alkylidene, which could explain the increased stereoselectivity in ROMP of NB, but apart from the X-ray structure of *syn*-**9** (dihedral angles N2-W1-C38-C39 15.4°, N2-W1-C38-H38 -164.5°) we do not have further experimental evidence, and other explanations for the effect are also possible. The reasons why we consider it still important to include these results in the catalytic examples are (i) their potential practical importance, and (ii) the fact that **5/9** is not the only MAP/MAP-adduct couple that we have found to display this behavior.

It is also highly relevant that the catalyst effects on the stereoselectivity of ROMP of norbornene-type substrates are very complex, and often cannot be fully interpreted simply by the model Prof. Schrock and Prof. Hoveyda have published in *J. Am. Chem. Soc.* **131**, 7962–7963 (2009), which primarily relies on the type of the ligand environment (bisalkoxide or MAP), and the relative size of the alkoxide vs. imido ligands to explain structure-selectivity relationships. As an example, in a comparative study performed under the same conditions, we observed 91/9 *cis/trans* and 23/77 *meso/racemo* ratios for W(NAr^{diiiPr})(neophylidene) (Me₂Pyr)(OCPh(CF₃)₂) (**5**), while 85/15 *cis/trans* and 67/33 (!) *meso/racemo* ratios for the corresponding molybdenum catalyst Mo(NAr^{diiiPr})(neophylidene) (Me₂Pyr)(OCPh(CF₃)₂), indicating a vast impact of the central atom on the stereoselectivity.

In relation to the catalytic examples, in general, we should point out that the topic of our communication is a synthetic approach towards Schrock catalyst adducts with increased air-stability

and retained activity. The catalytic examples primarily serve to demonstrate the retained activity, and specify the most typical and/or important differences one could expect upon using an adduct instead of its parent complex. Presenting in-depth comparative and mechanistic catalytic studies has not been our goal and simply does not fit the scope of the paper.

Finally, we would like to thank you again for your efforts and valuable remarks. The changes made upon your comments and proposals certainly have improved the manuscript, and we hope that the revised version will now be found suitable for publication in *Communications Chemistry*.

Sincerely,

Henrik Gulyás, PhD

senior research scientist, project leader, laboratory head

XiMo Hungary Kft.

H-1045 Budapest, Berliini utca 47-49, Berliini Park
43007 Tarragona, Av. Paisos Catalans 18, Fundació URV, CTTi
Cell phone: (+36)-30-7227-486; (+34)-644-749-303
Tel.: (+34)-977-55-8689; (+34)-977-55-8641; (+34)-977-55-8682

REVIEWERS' COMMENTS:

Reviewer #1 (Remarks to the Author):

The revised version of the manuscript satisfactorily addresses all the points raised by this reviewer. Reviewer #3, however, observed the need for a justification regarding the catalyst choice in metathesis reactions. That is an important point, and the reason for not doing this (as explained in the rebuttal letter) should be briefly discussed in the article. A sentence, such as the following, could fit on page 14 after line 345:

"Although an extensive catalyst and substrate screening is beyond the scope of this work, a few comparative metathesis studies will highlight the similarities and differences in the catalytic behavior of the adducts compared to that of the parent complexes."

In addition, it should be noticed that the first author holds a patent (EP3268377B1, WO2016142849A1), which includes in the claims the synthesis of the new alkylidene complexes described in the manuscript. He also belongs to the company XiMo Hungary Ltd. As emphasized in the added lines 45–46 on page 2, this company already commercializes olefin metathesis catalysts that are physically protected from air. This fact might constitute a competing financial interest. That being said, there are no signs that data, analysis or conclusions are lacking objectivity or integrity.

Since I already supported publication of the previous version, I confirm my recommendation to publish this paper in its present form.

Reviewer #2 (Remarks to the Author):

The Authors explained all my questions and followed my suggestions adding to the manuscript some fragments that are improving it. Although I am still seeing the roots of this work in Fürstner's old work, I think that now, the paper shall be published in CC. Definitely, this is a very solid piece of work.

Reviewer #3 (Remarks to the Author):

See attached file

In the revised version, manuscript **COMMSCHEM-20-0387A**, Henrik Gulyás et al. did not include any of my three remarks. I must express that I feel that the manuscript should be published in Communications Chemistry, but I must admit that the fact that authors did not succeed in answering any of my points makes me hesitate a little bit about the scope of the proposed approach and how difficult may be the development of new catalyst – based adducts. In this context, in their rebuttal letter, they argue that air stable 18e Mo MAP catalyst can be synthesized and they provide the stability constant of $\text{Mo}(\text{NAr}^{\text{diiPr}})(\text{neophylidene})(\text{Me}_2\text{Pyr})(\text{OCPh}(\text{CF}_3)_2)(2,2'\text{-bipyridine})$, which is almost two order of magnitude smaller than that of the W analogue. I assume that this so small stability constant, prevents obtaining the corresponding 18e- complex, thus reinforcing my concern that it is not clear if Mo MAP complexes are acid enough to be stabilized with this approach. In this context, I wonder about the extent of the following sentence of the manuscript “the present approach can readily be applied to molybdenum-alkylidenes, as well.” Is the approach only limited to the most acidic Mo complexes?

Regarding my second point, I agree with the authors that performing all reactions with all complexes is not necessary for the purpose of the present manuscript. However, I disagree with them that performing at least one catalytic test with **8** and **10**, would not provide any new insight. Indeed, these tests will show that **8** and **10** are able to catalyze olefin metathesis reactions. I assume that this is probably the case, but in the present version of the manuscript there is not a single example confirming this point.

Overall, I must admit that I am disappointed with how authors faced my concerns as I really think that including one air stable Mo MAP complex and illustrating that all synthesized pre-catalysts are active will improve the manuscript. However, I could accept the publication of the manuscript even in its present form if the two minor typos I have found are corrected.

Figure 3C is not a local maximum but a transition state.

Page 4 line 92, In my opinion **4** is not a W(VI)-bisalkoxide but a W(VI)-bisiloxide

Dear Reviewers,

We thank you for your final comments and the acceptance of our manuscript. We have found all of your additional observations important, and the manuscript has been modified further in accordance with each specific proposal.

We are truly grateful for the time and effort you have invested in examining, evaluating, and improving the manuscript. Should you wish to discuss further any aspect of the work in the future, you are most welcome to contact us.

Sincerely, on behalf of all authors,

Henrik Gulyás, PhD

senior research scientist, project leader, laboratory head